# Lahaul–Zanskar–Sham Valley Corridor in Indian Trans Himalayan Region Facilitates Dispersal and Gene Flow in Himalayan Ibex

**DOI:** 10.3390/biology12030382

**Published:** 2023-02-28

**Authors:** Gul Jabin, Stanzin Dolker, Bheem Dutt Joshi, Sujeet Kumar Singh, Kailash Chandra, Lalit Kumar Sharma, Mukesh Thakur

**Affiliations:** 1Zoological Survey of India, New Alipore, Kolkata 700053, West Bengal, India; 2Department of Zoology, University of Calcutta, Kolkata 700019, West Bengal, India

**Keywords:** biological corridors, landscape genetics, *Capra sibirica*, genetic diversity, connectivity

## Abstract

**Simple Summary:**

Wildlife corridors play a pivotal role to support free ranging wildlife in rapid changing landscapes which remains vulnerable due to growing human population, together with unplanned urbanization, industrialization, developmental activities and the inverse impacts of climate change. This triggers for undertaking urgent actions to not only ensure their long-term survival, but also securing sustainable socio-economic and environmental development. Using landscape genetics approach, we in the present study examined the movement and dispersal patterns of a wild mountain ungulate, Himalayan ibex which is considered an ecological engineer and also imparts a major prey base to the large carnivores in Indian trans-Himalayan Region. We identified the Lahaul–Zanskar–Sham valley (L–Z–SV) corridor supports the bidirectional movement of Himalayan ibex, and suggest science driven management interventions for the conservation of Himalayan ibex and other sympatric ungulates in the trans-Himalayan region.

**Abstract:**

Wildlife corridors that connect mosaic habitats in heterogeneous mountainous landscapes can be of high significance as they facilitate the genetic and demographic stability of free-ranging populations. Peripheral populations of widespread species are usually ignored in conservation planning. However, these populations retain locally common alleles and are genetic reservoir under the changing climatic conditions. *Capra sibirica* has widespread distribution, and its southern peripheral population is distributed in the Indian trans-Himalayan region (ITR). In the present study, we studied the spatial distribution and genetic make-up of Himalayan ibex from the ITR following the landscape genetics approach. We obtained 16 haplotypes at the mitochondrial d-loop region and found a stable demography in the past with a recent decline. With 10 nuclear microsatellites, we ascertained 111 unique individuals assigned into two clusters following Bayesian and non-Bayesian clustering analysis with several admixed individuals. We also recorded 25 first-generation migrants that reflected relatively high dispersal and gene-flow across the range. We identified a 19,835 sq.km suitable area with 13,311 sq.km in Ladakh and 6524 sq.km in Lahaul-Spiti. We identified a novel movement corridor for Himalayan ibex across the Lahaul–Zanskar–Sham valley (L–Z–SV) that displayed a fairly good conductance with low genetic divergence among the samples collected on the L–Z–SV corridor. We propose declaring a protected area in the Lahaul and Kargil districts to prioritize dedicated conservation efforts for the Himalayan ibex and other sympatric ungulates that impart a major role in the diet of large carnivore and balancing ecosystem services in the trans-Himalayan region.

## 1. Introduction

Biological corridors connect large landscapes and facilitate the movement of wildlife for various purposes [1,2,3]. The corridors also help in maintaining the genetic diversity and demographic stability of wildlife and can also minimize the ill effects of genetic inbreeding, demographic stochasticity and genetic bottleneck [3,4,5,6].

Recently, the Convention on Biological Diversity (CBD) proposes a post-2020 global biodiversity framework where monitoring and maintaining genetic diversity within all species (domestic and wild) is prioritized under the Goals and Action Targets (Convention on Biological Diversity, 2022). In light of this, we believe that the landscape genetics approach bridges the ecological information and population genetic estimates to spatially represent genetic diversity patterns, as well as the functionality of the biological corridors [7,8,9,10]. Any potential habitat loss and fragmentation often restrict the movement of highly mobile species and limit their gene flow in the landscape [11,12]. Thus, it is vital to secure long-term functional connectivity across wide landscapes to support a number of species which remain connected through an ecosystem cascade and are otherwise difficult to monitor [7,13]. Furthermore, inevitable anthropogenic dependency and harvesting the natural resources decline the suitable habitats, which needs to be prioritized with spatial planning to support the species’ movement [3,5,6,14,15,16].

Siberian ibex (*Capra sibirica*) is the largest species of ibex in the genus “*Capra*” which occupies mountains, deserts and plains of many Asian countries (China, Mongolia, Russia, Kazakhstan, Kyrgyzstan, Kazakhstan, Tajikistan, Pakistan, Afghanistan and India) [17,18]. The Siberian ibex in India (hereafter referred to as Himalayan ibex) exhibited distinct morphological features [18] and formed a distinct phylogenetic lineage [19]. In India, the Himalayan ibex is restricted to the Ladakh as well as Lahaul and Spiti districts of Himachal Pradesh, which is a part of the trans-Himalayan region of India located at the western edge of the Indian Himalayan region (IHR) [19,20,21]. The limited habitat occupied by the Himalayan ibex makes it vulnerable to habitat disruption, urbanization, climate change and other anthropogenic activities. As per IUCN Red list, the Siberian ibex is listed as “Near threatened” [17] due to a decline in population worldwide and in India, and it is listed as Schedule I under the Wildlife (Protection) Act, 1972. The trans-Himalayan region is home to many endangered species such as the snow leopard, the Himalayan brown bear, the Himalayan wolf, the Ladakh urial, the Himalayan ibex, etc. [14,21,22], and the genetic assessment of wildlife from this region is very limited [19,23,24]. In the present study, we aimed to understand the spatial genetic patterns of the Himalayan ibex and assess the functional connectivity in the trans-Himalayan region that supports the movement and gene flow among ibex populations following landscape genetics approaches.

## 2. Materials and Methods

### 2.1. Study Area

In India, the Himalayan ibex is distributed in Ladakh and Himachal Pradesh. The study was carried out in the following areas—Ladakh (LA): Leh district (Changthang, Nubra and Sham) and Kargil district (Drass and Zanskar) while in Himachal Pradesh it was carried out in the Lahaul and Spiti district (LS) [19,20], falling under the trans-Himalayan region extending from 31–33° N latitudes to 75–80° E longitudes (Figure 1). The study area covers a total of 106,014 km^2^ (86,983 km^2^ of Ladakh and 19,031 km^2^ of Lahaul-Spiti) with varying elevation from 2400 to 6000 m, exposed to extreme harsh climatic conditions with cold temperature, little rainfall and reduced oxygen levels with high levels of UV radiation. This region also represents arid and xeric conditions, which supports very low vegetation; however, it harbors a varied range of threatened flora and fauna.

### 2.2. Occurrence Data

Field surveys were undertaken during the 2018–2021 period. We first stratified the study landscape based on the forest types and topography to cover all the logistically possible habitat patches in the reported distribution range of ibex in trans-Himalayas. In order to record the species’ presence, we adopted a three-pronged approach, i.e., transects/trail surveys, questionnaire surveys and also the remote camera traps following Joshi et al.’s work [25]. Since it was not logistically feasible to cover the entire area systematically due to rugged terrain, unpredictable weather, poor resources and other constrains, we carried out representative sampling in the trans-Himalayan region. We recorded 406 presence points of Himalayan ibex based on the cumulative efforts of 384 sign surveys (47 direct sighting and 337 pellet samples) and 22 records from 24 camera traps. We undertook field surveys in Eastern Ladakh (Changthang), but did not find any sign of ibex presence (Figure 1). Finally, we used 213 spatially independent locations for habitat suitability in species distribution modeling.

### 2.3. Sample Collection, PCR and Sequencing

A total 337 pellet samples, i.e., 148 from LS and 189 from LA region, were collected and stored in collection vials with silica (Figure 1). We opportunistically found 17 tissue samples, i.e., one from LS and 16 from LA, which were also collected and stored in 90% alcohol. The outer layer from each pellet sample was scraped and genomic DNA was extracted using QIAmp Fast DNA Stool Mini Kit (QIAGEN, Hilden, Germany) following manufacturer’s instructions, while DNA from tissue samples was extracted using the phenol–chloroform–isoamyl method [26]. The quality of the extracted genomic DNA was verified on 1% agarose gel. The PCR amplifications were performed in 10 μL reaction volume comprising approximately 5–10 ng of template DNA, 1U Taq polymerase (Takara), 10X PCR buffer, 2 mM MgCl_2_, 2.5 mM dNTPs mix, 0.1 µM of each primer and 0.1 µg/µL BSA on Veriti thermal cycler (Applied Biosystems, Waltham, MA, USA). We amplified a partial fragment of mitochondrial gene Cytb [27] and synthesized a novel primer set to amplify d-loop region (F-5′CCCATGTAATACAGACATATAGTCCTCA, R-5′GCGGCATGGTCAACAAGCTCG TGATCT) for species confirmation. The thermal cycling conditions were as follows: initial denaturation at 95 °C for 2 min followed by 40 cycles of denaturation at 95 °C for 30 s, annealing at 53 °C for 1 min and extension at 72 °C for 45 s with final extension at 72 °C for 10 min. Sanger sequencing was performed on Genetic analyzer 3730 (Applied Biosystems, Foster City, CA, USA). The generated sequences were cleaned manually using Sequencher software v5.4.6 (Gene Codes Corporation, Ann Arbor, MI, USA) and subjected to NCBI-BLAST for similarity search.

### 2.4. Microsatellite Genotyping

Genetically confirmed samples were then attempted for microsatellite genotyping. We first tested 22 bovid and cervid microsatellites for positive amplification (Appendix A) [28,29], out of which only 16 markers amplified successfully. Each marker was genotyped with four reference tissue samples (Appendix A) and the loci that demonstrated a relatively high success rate ≥ 85% were used in generating four multiplexes considering the dye chemistry and amplicon size. PCRs were undertaken on Veriti thermal cycler using Qiagen Multiplex PCR kit following manufacturer’s instructions. Amplified products were resolved by capillary electrophoresis on Genetic analyzer 3730 (Applied Biosystems, Foster City, CA, USA) and alleles were scored manually on Gene Mapper 4.1 (Applied Biosystems, Foster City, CA, USA).

### 2.5. Distribution Modelling

Identification of species-specific suitable habitat in the study area was undertaken based on Maximum Entropy algorithm in the software MaxEnt V.3.4.3 [30]. Initially, we used 12 environment variables and 19 bioclimatic variables downloaded from WorldClim database (http://www.worldclim.org/bioclim accessed on 24 April 2022) (Appendix A). Topographic variables (elevation, slope and aspect) were extracted from the digital elevation model and Euclidian distance from rivers and roads was calculated in ArcGIS 10.8 software (ESRI 2019). MODIS satellite data were downloaded for Land Use Land Cover (LULC) classification and then the area was classified into various LULC classes such as snow cover, barren, grassland, water, human-dominated area, forest, etc. The Human Influence Index was downloaded from Socioeconomic Data and Applications Centre database (SEDAC—https://sedac.ciesin.columbia.edu accessed on 25 April 2022). Each variable raster was re-sampled at 1 km resolution and converted into ascii format in ArcMap 10.8 software (ESRI 2019) to bring uniformity in the data. Based on the Pearson correlation coefficients (r) threshold of 0.8, highly correlated variables were omitted from the analysis (ENM tool-R package). Finally, we selected 17 variables for species distribution modelling of ibex (Appendix A). A bias file was prepared and a best fit model was determined based on the lowest Akaike’s information criteria (AICc) value in ENM-eval package of R [31]. For modelling evaluation, we used 70% of locations as training data and the remaining 30% as test data to estimate the statistical significance. All the variables’ relative and absolute contribution to the model were considered individually by Jack-knife test to assess the model accuracy. The modelling was assessed following receiver operating characteristics (ROCs) area under the curve (AUC) with a threshold value higher than 0.8 < AUC < 1, which provided a good prediction model.

### 2.6. Sequence Data Analysis

All generated sequences were aligned on MEGA X [32] using Clustal W tool and trimmed to generate a similar length of sequences for genetic analysis. We used cytb sequences for species determination and d-loop sequences for population genetics analysis due to their hyper-variability nature. Haplotype diversity indices were calculated using DnaSP version 6.12.3 [33]. Neutrality tests, i.e., Tajima’s D, Fu and Li’s F and D were also carried out to evaluate the demographic effects using program DnaSP version 6.12.3 [33]. The demographic pattern of the population across time was reconstructed through the coalescent Bayesian skyline method [34] which provides estimates of changes in the effective population size over time following the program BEAST version 2.6.3 [35]. The molecular evolution was calibrated assuming a substitution rate of 0.27 × 10^−7^ substitutions/site/year following Tarekegn et al. (2018) [36] for d-loop, and the results were visualized using Tracer 1.7.1 [37]. A median-joining network was constructed using the NETWORK software [38] and the generated haplotype was then plotted onto the map to understand the sharing of the haplotype across the range in ArcMap 10.8 (ESRI 2019).

### 2.7. Microsatellite Data Analysis

We repeated genotyping at least three times with a large fraction of samples and heterozygotes were ascertained with two independent runs. Then, we calculated genotype errors such as allele drop out (ADO) and false allele (FA) using PEDANT v 1.0 involving 10,000 search steps for enumeration of per-allele error rates using the repeat genotypes. The PIC value, an indicator of marker’s informativeness and null allele frequencies, was calculated using CERVUS version 3.0 [39]. Since increasing the number of loci may increase the chance of obtaining a unique genotype and in non-invasive samples, it may overestimate the individuals in a population through genotypic error. We prioritized a panel of seven loci for individual identification based on their least or no genotyping error, high amplification success (≥80%) and a high discriminating power. We identified unique genotypes using GenAlEx version 6.51 [40] and the individual assignment was conducted using GeneCap version 1.2.2 [41]. The locus wise and cumulative probability of identity for unrelated individuals (P_ID_) and siblings (P_ID_ sibs) was calculated using the identity analysis module in GenAlEx version 6.51 [40].

Furthermore, the population genetic analysis was attempted using ten microsatellite markers with identified unique genotypes having > 70% success rate. The per-locus diversity was quantified by estimating the numbers of observed (N_a_) and effective alleles (N_e_), as well as observed (H_o_) and expected (H_e_) heterozygosity using GenAlEx version 6.51 [40]. For the Hardy–Weinberg equilibrium estimation, we followed the probability test approach [42] using the program FSTAT version 2.9.4 [43]. The unbiased estimator of inbreeding coefficient F_IS_ was calculated according to Weir and Cockerham (1984) [44] using FSTAT version 2.9.4 [43]. Linkage disequilibrium (LD) was tested using GENEPOP version 4.7 to determine the extent of distortion from independent segregation of loci following 10,000 dememorizations, 100 batches and 10,000 iterations per batch [45].

The population’s genetic structure was attempted using three spatially and non-spatially explicit Bayesian and non-Bayesian clustering methods. In the Bayesian clustering method, we run STRUCTURE software v.2.3.4 with 500,000 MCMC and 50,000 burn-in with 20 runs at each K. The appropriate *K* value was determined by calculating ad hoc quantity (Δ*K*) following Evanno et al.’s 2005 work [46]. Following the previous studies [47,48,49,50], each individual was assigned to the inferred clusters using a threshold proportion of membership (*q*), i.e., *q* ≥ 0.80; otherwise, an individual was determined as admixed if the *q* value was less than 0.80.

The second spatial Bayesian clustering was performed using GENELAND, which takes in account both the genotype data and geo-referenced information. We allowed K to vary from 1 to 10 with both correlated and uncorrelated allele frequency models. We conducted 20 independent runs with 1,000,000 iterations and a thinning of 1000 while other parameters were kept as default. The top run with the highest posterior probability algorithm was selected and post-processing was performed using 100 × 100 pixels in spatial domain with a 100 burn-in period.

In the non-Bayesian clustering method, we used DAPC in the “adegenet” package [51] of R (R 4.0.3, R Core Team, 2022) and retained the first 20 PCs after cross-validation. Furthermore, to evaluate the migration rate between the LA and LS population, we ran BAYESASS at 9 × 10^6^ MCMC iterations [52], after which 10^6^ were discarded as burn-in and sampling frequency was conducted every 2000 intervals. Runs were carried out at a migration rate prior to 0.05 while other parameters were kept at default. We used GENECLASS2 [53] to detect first-generation migrants using the likelihood estimator L_h_/L_max_ with algorithm as described by Paetkau et al. (2004) [54] and the assignment threshold was set at 0.01. The significant assignment of migrants in GENECLASS 2 [53] was contingent based on *p* < 0.01.

In order to evaluate the relationship between the genetic and geographic distances, isolation by distance (IBD) analysis was conducted using Allele in Space (AIS) [55], following the Mantel test. A genetic landscape based on pairwise population genetic divergence was created to understand the genetic divergence between the populations in the landscape. A genetic distance estimate pairwise Fst was obtained from GenAlex using the microsatellite data and a genetic divergence model was generated as raster surface using the single-species genetic divergence package of the Genetic Landscape GIS toolbox V.10.1 in ArcMap 10.8 (ESRI 2019).

### 2.8. Corridor Connectivity Using Landscape Genetics

We evaluated the current connectivity among the habitats of the Himalayan ibex in the landscape based on the circuit theory as implemented in the Circuitscape software V.4.0 [56] following its proven applicability to model movement and gene flow in plants and animals [7,56,57]. We constructed the conductance model with the raster of habitat suitability as the base and established possible connections by assessing the conductance value. In order to avoid over-prediction, we ratified the occurrence points with a 5 km distance on the basis of ibex home range [58,59] and the mid-points between each point were used as nodes to predict the connectivity model where the higher value of conductance indicated the possible routes of ibex movement across the landscape.

## 3. Results

We found a 19,835 sq.km area suitable from the 106,014 sq.km area for the Himalayan ibex, which comprises 13,311 sq.km in Ladakh and 6524 sq.km in Lahaul-Spiti (Figure 2). The AUC of the model was 0.906 ± 0.008 SD, indicating the accuracy and relevance of the selected variables in predicting the suitable habitat. The Jackknife test revealed that bioclim 17 (Precipitation of Driest Quarter) was the most important variable (32.1%) followed by precipitation seasonality (21.4%), elevation (9.6%) and distance to grassland which was the least important variable (0.9%), while the distance to grassland variable exhibited the highest contribution (29.3%), followed by bioclim 17 (20.9%), elevation (12.3%) and the topographic wetness index which was the least contributing variable (0.6%) (Appendix A). The generated model revealed that much of the suitable areas in the Leh and Lahaul region under fell into the non-protected areas (Figure 2).

Of the 337 pellet samples, 148 samples were identified as ibex, of which 126 samples were amplified with 10 microsatellite markers. We obtained 16 haplotypes characterized by 34 substitutions with an average of 6.977 nucleotide differences in the control region (Appendix A) with three haplotypes H1, H2 and H6 shared between both the LA and LS regions (Figure 3A and Appendix A). The overall haplotype diversity (Hd) and nucleotide diversity (π) were 0.907 ± 0.019 and 0.0206 ± 0.0025 (Appendix A), respectively. The network displayed shared haplotypes as well as unique haplotypes in both LA and LS (Figure 3A). The mismatch modelling appeared as multimodal, which explained why the past stable effective population and Bayesian skyline plot also showed a stable past population with a recent decline (Figure 3B,C).

We found 10 loci with ≥80% success rate and identified 111 individuals (60 from LA and 51 from LS) with a select panel of seven microsatellite loci (cumulative P_ID_ of 3.2 × 10^−8^ and cumulative P_ID_ sibs of 1.4 × 10^−3^). Locus-wise P_ID_ and P_ID_ sibs ranged from 0.05 to 0.1 and from 0.35 to 0.46, respectively (Table 1).

We found a genotyping error, i.e., ADO, which ranged from 0 to 0.2189 while no potential FA was detected in any of the examined loci (Table 1). All loci were polymorphic with a PIC value ≥ 0.5, except ETH10, and all loci except for CSSM14 deviated significantly from HWE. The mean Na and Ne were 12.30 ± 0.831 and 3.849 ± 0.331, respectively, while Ho and He were 0.426 ± 0.051 and 0.718 ± 0.031, respectively (Table 1). The population-wise diversity estimates are given in the Appendix A. The overall mean inbreeding coefficient was 0.393 ± 0.060, which showed a moderate level of inbreeding in the population (Table 1).

We obtained two possible clusters in STRUCTURE (Appendix A) based on delta K plot. The majority of the samples, i.e., 56.75% collected from LA and LS were accordingly assigned into two clusters, while two individuals from LS were assigned with LA and eight individuals sampled from LA were genetically assigned with the LS population (Figure 4A). We obtained 48 admixed individuals, not assigned to any of these clusters, indicating a bidirectional gene flow between LA and LS. DAPC also resulted in a similar output and showed intermixing between two clusters (Figure 4B and Appendix A). We interpret gene–land results with the correlated allele model, which showed four meta-populations where most of the individuals were assigned into two major clusters, i.e., Cluster II and IV (Figure 4C,D).

The BayesAss program showed a continuous and bi-directional movement of individuals from either side supporting gene flow with a significant and symmetric migration of 5.4% from LA to LS and of 4.7% from LS to LA. Within population migration, 94.5% were in LA and 95.2% in LS (Appendix A). We also identified 25 first-generation migrants by Geneclass (17 individuals captured in LS from LA and 8 individuals captured in LA from LS) (Figure 5). These first-generation migrants displayed significant cross-assignment based on the log (L_h_/L_max_) value (*p* > 0.01) (Appendix A). The Mantel test nullified the IBD concept and showed a weak relationship between the geographic distance on the genetic distance with R = 0.1369 (Appendix A).

In the landscape connectivity model, we obtained high intensity current flow from the Spiti valley to Lahaul and from Lahaul to Zanskar to Sham valley in a vertical manner. This indicated that the study area represents a good physical connectivity (Figure 5A), and we identified a novel corridor, the Lahaul–Zanskar–Sham valley (L-Z-SV) that connect LA to LS. While investigating genetic divergence among samples over the habitat suitability model with the ensemble approach, we obtained a relatively high genetic divergence in Spiti valley and Drass while the predicted corridor from Lahaul to Zanskar to Sham valley exhibited a relatively low genetic divergence (Figure 5B). This corroborated the fact that the ibex is likely to utilize the predicted novel corridor “L-Z-SV” effectively and verified the functionality of the corridor (Figure 5A).

## 4. Discussion

The genetic diversity estimates of the Himalayan ibex were in the range of other similar ungulates such as the Nubian ibex (*Capra nubiana*), Korean goral (*Naemorhedus caudatus*), Hangul (*Cervus hanglu hanglu*) and blue sheep (*Pseudois nayaur*) [60,61,62,63]. The neutrality tests were negative and supported population expansion in the past. The mismatch distribution curve indicated a population under demographic equilibrium and did not support a population bottleneck in the past. The BSP also revealed a stable demographic history over the 600 kya with an apparent decline in the past 30 kya (Figure 3B,C). Interestingly, only one haplotype, H2, was detected in the Spiti region showing one genetic variation in Spiti individuals, whereas samples collected from Leh, Kargil and Lahaul exhibited site-specific haplotypes as well as shared haplotypes, indicating a good gene flow in the study area. H6 was shared between Leh and Lahaul; H1 was shared between Lahaul and Kargil. We did not observe any shared haplotypes between Leh and Kargil, which may be due to the restricted gene flow between areas or due to the limited sampling. The combined results support the stable population at present with moderate diversity. However, these results should be interpreted with site-specific findings and we believe a follow-up and through sampling design will bring new insights in the future conservation and long-term monitoring of the Himalayan ibex in trans-Himalayas.

The population assignment showed two distinct clusters with 48 admixed individuals that reflected that the ibex in LS and LA is well-connected by means of the L–Z–SV corridor (Figure 2A, Figure 3A,B and Figure 5A). The ibex retains the ability to move widely and disperse across high mountains and is often sighted on the pinnacles of high mountain ridges. They travel at long distances and tough terrains where highly elevated mountains do not restrict their dispersal and gene flow [18,64,65]. The same was evident in the present study. We found climatic variables, especially precipitation of the driest quarter and precipitation seasonality, as important factors compared to other environmental variables for predicting ibex habitat suitability. This indicated that precipitation plausibly regulates the flora in the landscape and thus the diet and the habitat suitability of the ibex. Consequently, the ever-increasing temperature in the eco-sensitive habitats of the trans-Himalayas due to global warming may alter the habitat suitability and physiological responses of the ibex. The Himalayan ibex forms the peripheral population of the widely distributed Siberian ibex and Joshi et al. [19] also reported that it is an evolutionary distinct phylogenetic lineage of the Siberian ibex. Thus, we propose this unit or phylogenetically distinct lineage to be managed at a regional level. We identified a bi-directional gene flow among the populations of LA and LS through the novel corridor L–Z–SV, with sustenance due to the conducive environment and suitable habitat in this region. The average annual precipitation in Zanskar (~250 cm) and Lahaul (~1100 cm) (https://hplahaulspiti.nic.in/climate/ accessed on 15 December 2022) is higher than in Leh (~200 mm) [66,67,68,69,70,71] as well as Lahaul which receives higher snowfall as compared to the Spiti region. Westerlies play the main role for precipitation in the region and cause snowfall in the region as Leh is in the rain shadow area of the Tibetan Plateau. Additionally, Indian monsoon does not have much effect in Leh due to the Zanskar mountain range and can only cause drizzle in the Lahaul region as the Pir Panjal Mountains also act as a barrier for Lahaul. Therefore, we presume that L–Z–SV is more suitable for floral diversity due to a higher precipitation and therefore more conducive to ibex movement. Lahaul and Zanskar also have an abundant glacial system [69,72] which melts in summer, increasing alpine vegetation in the form of pastures, which is ideal for grazing. The river network system would be another factor making this corridor more conducive as all the three regions (L–Z–SV) are evolved along the side of rivers, i.e., Sham valley is present at the Indus riverside, Zanskar is evolved around the catchment area of Zanskar and Suru river, whereas Lahaul is adjoining to the Chandra river. The Sham valley, despite being present in the Leh region, would be more conducive than the other region of Leh due to warmer temperatures and river catchment areas, making it more productive also thanks to glaciers in few villages such as Domkhar [66], while the Eastern Ladakh mostly has barren plains, which is not preferred by the ibex. Another factor would be the higher plant diversity in this linking landscape as compared to other areas of Ladakh. The Siberian ibex is known as a mixed feeder as it feeds on a large variety of plants, its main diet including herbaceous plants such as Cyperaceae, Poaceae, Asteraceae and Rosaceae. Preferably, the ibex feeds on forbs sp., such as senecio, and avoids graminoids such as Kobresia, even though it is abundant [20,73]. The higher floral diversity and density in the Zanskar-Suru valley of the Ladakh and Lahaul region, as compared to the other regions considered in this study area may indicate that the corridor seems to provide an abundant diet, such as Asteraceae and Rosaceae family plants, which are highly present in these areas [74,75,76]. The habitat suitability model reveals ibex suitability mainly out of protected areas, especially in the Lahaul and Kargil regions, and thus, emphasis should be provided to turn these areas into protected areas. Since the ibex shares its habitat with other ungulates such as the blue sheep and the urial, as well as with its predators such as the snow leopard and the Himalayan wolf, turning these regions into protected areas will also help in the conservation of these species and in maintaining the food web in the trans-Himalayan landscape.

## 5. Conclusions

Through the present study, we suggested the management and protection of the predicted novel corridor, ‘‘L-Z-SV’’, from urbanization and fragmentation in order to facilitate gene flow. This is important because a decline in the connected population will result in a survival of the species in small and isolated patches, which in turn may result in the loss of genetic diversity and upraise in genetic inbreeding. The population at the periphery should also be given equal priority in conservation to maintain and protect the genetic diversity of the population from loss of genetic diversity. We recommend the long-term monitoring of the ibex population and other species’ diversity to prevent inbreeding and local extinction. It is imperative to develop a management policy between both states for community conservation of the predicted corridor and to declare a protected area in these regions.

## Figures and Tables

**Figure 1 biology-12-00382-f001:**
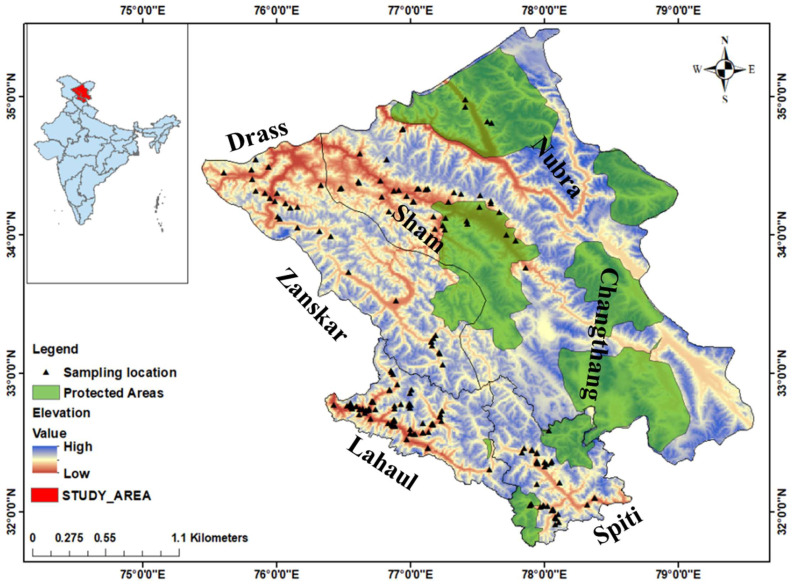
Geographic map of study area showing protected areas and sampling location of Himalayan ibex in distribution range of India. (LA—Nubra, Sham, Drass and Zanskar; LS—Lahau, Spiti).

**Figure 2 biology-12-00382-f002:**
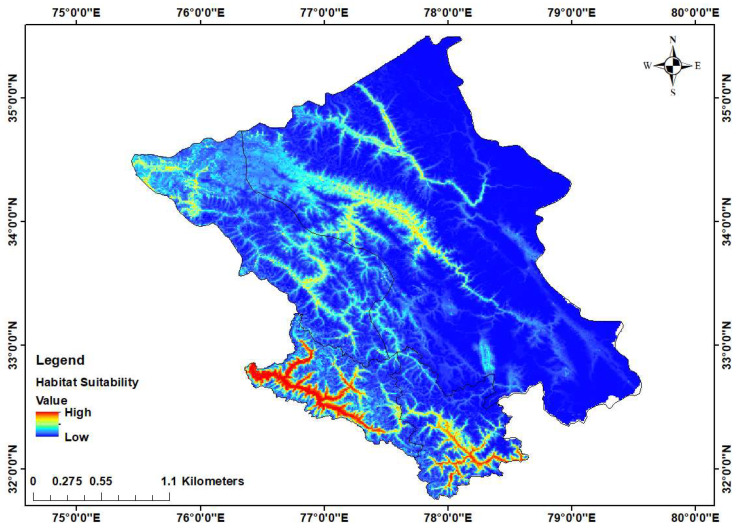
Habitat suitability map with higher suitability (red) and lowest suitability (dark blue) shows maximum suitability in Lahaul-Spiti region.

**Figure 3 biology-12-00382-f003:**
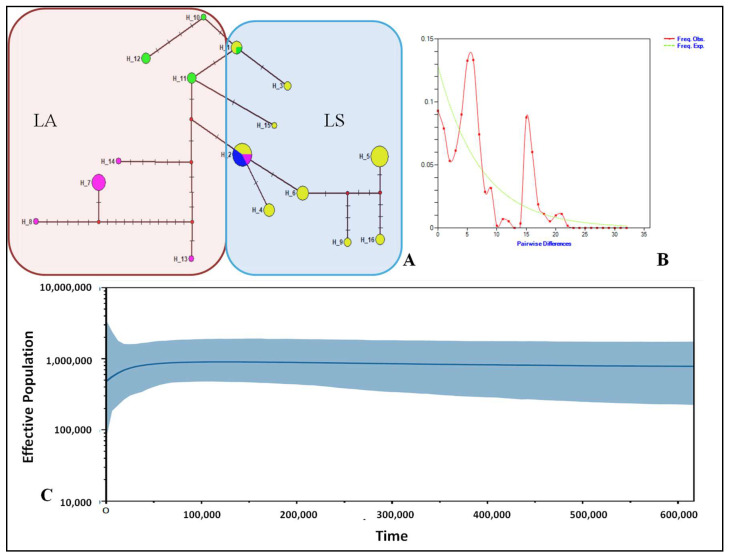
(**A**) Haplotype network generated using d-loop gene of ibex in Lahaul-Spiti and Ladakh regions. (**B**) Mismatch modelling curve shows multimodal explaining population equilibrium in past. (**C**) Bayesian skyline plot reveals stable population over last thousand years and shows decline in last ~30 kya.

**Figure 4 biology-12-00382-f004:**
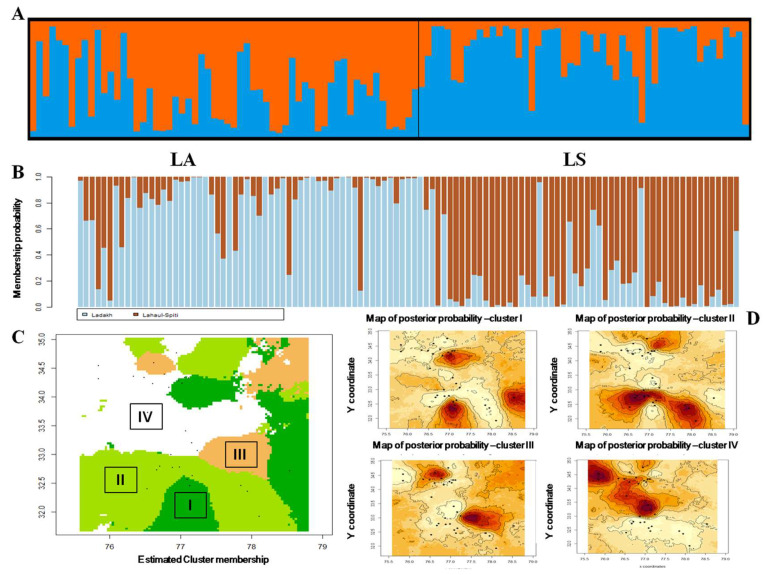
(**A**) Bayesian clustering using STRUCTURE at K-2 with ibex individuals (LA-60, LS-51). (**B**) Non-Bayesian clustering using DAPC with 111 ibex individuals. (**C**) Estimated cluster membership shows spatial distribution of the four inferred genetic clusters across the study areas in Geneland using ibex individuals (K = 4 gives the best run showing highest average posterior probability). (**D**) Maps of posterior probability for each inferred cluster show the spatial location of genetic discontinuities of clusters I, II, III and IV, whereas sample locations are represented by black dots.

**Figure 5 biology-12-00382-f005:**
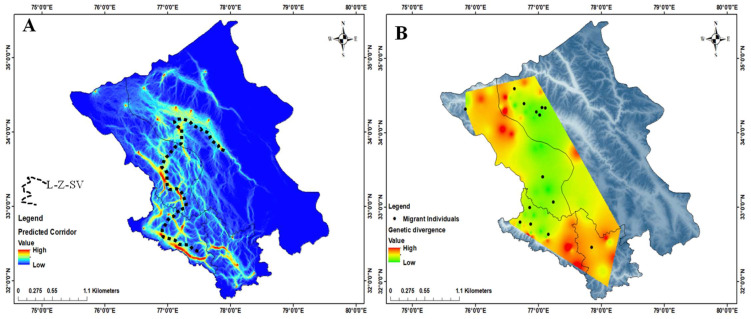
(**A**) Predictive corridor map for movement of ibex in the study area “L-Z-SV corridor”. (**B**) Genetic divergence plotted across the landscape shows low divergence in the central region (L–Z–SV corridor) and the locations of migrants plotted on the map showing movement in the corridor landscape.

**Table 1 biology-12-00382-t001:** Population genetic indices for the Himalayan ibex in the Indian trans-Himalayan region.

Locus	N	Na	Ne	Ho	He	uHe	PID by Locus	PID Sibs by Locus	Cum. PID	Cum. PID Sibs	F	F_IS_	PIC	ADO	FA
Haut14 ^†^	102	15	5.555	0.333	0.820	0.824	5.2 × 10^−2^	3.5 × 10^−1^	5.2 × 10^−2^	3.5 × 10^−1^	0.593	0.584	0.801	0.089	0.00
ETH152 ^†^	97	18	5.171	0.526	0.807	0.811	5.7 × 10^−2^	3.6 × 10^−1^	2.9 × 10^−3^	1.3 × 10^−1^	0.348	0.360	0.785	0.0156	0.00
BM415 ^†^	110	11	4.213	0.491	0.763	0.766	8.9 × 10^−2^	3.9 × 10^−1^	2.6 × 10^−4^	5.0 × 10^−2^	0.356	0.328	0.730	0.125	0.00
CSSM14 ^†^	102	11	4.156	0.745	0.759	0.763	9.1 × 10^−2^	3.9 × 10^−1^	2.4 × 10^−5^	2.0 × 10^−2^	0.019	0.016	0.726	0.015	0.00
ETH225 ^†^	90	10	3.989	0.433	0.749	0.754	9.8 × 10^−2^	4.0 × 10^−1^	2.4 × 10^−6^	7.9 × 10^−3^	0.422	0.405	0.714	0.096	0.00
INRA35 ^†^	89	10	3.562	0.494	0.719	0.723	1.2 × 10^−1^	4.2 × 10^−1^	2.8 × 10^−7^	3.3 × 10^−3^	0.313	0.276	0.680	0.000	0.10
BM1824 ^†^	104	12	3.511	0.250	0.715	0.719	1.2 × 10^−1^	4.2 × 10^−1^	3.3 × 10^−8^	1.4 × 10^−3^	0.650	0.643	0.679	0.022	0.00
CSRP6	106	12	3.489	0.406	0.713	0.717	1.3 × 10^−1^	4.2 × 10^−1^	4.1 × 10^−9^	5.9 × 10^−4^	0.431	0.405	0.670	0.0306	0.10
CSSM19	96	14	2.962	0.417	0.662	0.666	1.5 × 10^−1^	4.6 × 10^−1^	6.1 × 10^−10^	2.7 × 10^−4^	0.371	0.361	0.628	0.031	0.00
ETH10	105	10	1.883	0.162	0.469	0.471	3.0 × 10^−1^	5.9 × 10^−1^	1.9 × 10^−10^	1.6 × 10^−4^	0.655	0.644	0.448	0.033	0.00
Mean	100.1	12.30	3.849	0.426	0.718	0.721					0.416	0.393			
SE	2.188	0.831	0.331	0.051	0.031	0.031					0.060	0.060			

^†^: Markers used for individual identification; N: No. of alleles; Na: No. of observed alleles; Ne: No. of effective alleles; Ho: Observed heterozygosity; He: Expected heterozygosity; uHe: Unbiased expected heterozygosity; F: Fixation index; PID: Probability of identity; F_IS_: Overall inbreeding; ADO: Allele drop out; FA: False allele.

## Data Availability

All the sequences generated were uploaded on NCBI/GenBank database (d-loop: OQ378582-OQ378597 and cytb: OQ378598-OQ378601).

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
