# Peer review of "Lahaul–Zanskar–Sham Valley Corridor in Indian Trans Himalayan Region Facilitates Dispersal and Gene Flow in Himalayan Ibex"

_biology, 2023, doi:10.3390/biology12030382_

Round 1
Reviewer 1 Report
I've read the manuscript entitled "Lahaul-Zanskar-Sham valley corridor in Indian Trans Himalayan region facilitated dispersal and gene flow in Himalayan ibex" by Gul Jabin and colleagues.
The topic is relevant and the manuscript has a good potential, the authors have conducted an impressive amount of different analysis and provide interesting results. However, some major points needs to be addressed before considering it for publication.
General comments:
The text (particularly the introduction and discussion sections) needs deep restructuring. The topics are introduced in a superficial way and are not always well connected one to the other (e.g., - but there are examples of that - lines 45-46). Given some of the methodological issues, discussion should provide some warning about the interpretation of the results and some limitations of the study should be explicited.
In addition, several typos are present and the writing is not accurate, punctuation is often missing or misplaced and sentences are too long and difficult to follow. I think the authors should pay more attention to that and spend some time proof-reading the manuscript before submission, that would help reviewer to actually and more easily evaluate the scientific quality of the work.
The methods and results also would benefit from a careful revision, especially of punctuation.
Figures axis and legends are often not well readable (text too small and often "squeezed"). Figure 1 should be discorporated in two figures, a first one showing the sampling points, that can be introduced in the methods, and a second one with the other three maps that shows the results. In addition, at least in the first map, names and/or acronyms of the localities that are mentioned in the text needs to be provided. As it is now, unless a reader knows already the area very well, is nearly impossible to follow the explanations provided by the authors about movements and corridors.
Another major concern I have is related to the quality of the DNA extracted from sun dried faecal samples. As you perform several different analysis based on that, it would be nice to have some more information about the checks you have done for DNA quality. Herbivores faecal samples are known to yeld low quality DNA (see for example Willish et al., 2012 https://link.springer.com/article/10.1007/s10682-011-9486-6) so a bit more of attention should be placed on that. For example, did you do repeated genotyping of any of the samples to build consensus genotype with msats? How did you calculate dropout and FA rates without repeats?
In addition: you wrote that samples were sun dried. Why did you chose this protocol? UV lights would destroy the small amount of DNA present on the faeces surface thus further reducing DNA quality and quantity. For this reason, is even more surprising that you were able to extract enough DNA and of enough quality to perform all the analysis you present. It would be very important that you address this point in the text.
Some other details are not clear: you selected 10 microsatellites but then only used 7 of them. It is not clear why and how did you select them.
Finally, about the MAX-ENT analysis, you did not provide any detail about your sampling design. Was it opportunistic? Did you perform transects? Was the sampling effort homogeneous in the whole area? How were camera trap positioned? How did you treat data? (e.g., how did you account for possible double counts of the same individuals). More details on that should be provided in order to ensure that the data are suitable for the analysis that you then present.
Detailed comments:
As already stated before, the text needs careful revision. Below you can find some comments but they are not exhaustive, especially about language and typos. Please carefully check the language yourself.
Introduction:
line 37: I would change it with “Biological corridors connect”.
Line 39-40: what does genetic stability means? I would change the expression “alleviate”
line 54: Siberian ibex (no capital letter, here and throughout the text). In addition, if you want to refer to the Siberian ibex in your study site as Himalayan ibex I would introduce it here, and explain why (as you currently do in the discussion).
line 58: to what the brackets (hereafter; Himalayan ibex) refers to? I don't think it can refer to the mentioned area? I guess it refers to the species? Should be placed in the correct place (see previous comment).
Line 60: what do you mean with “limited habitat”?
Line 69: not sure “finite” is the correct word.
Line 71: “aims at understanding”.
Figure 1: Panels B C and D report results and should therefore pre presented in the appropriate session. Legend and axis texts are nearly impossible to read. In addition, please add to the map names or acronyms of the regions/areas/localities you then mention in the text, otherwise it is impossible for a reader to identify them.
Material and methods
(should be session 2, not 1 as 1 is already the introduction, please re number everything correctly).
Sample collection. I'm really surprised that sun dried faecal pellets provided enough DNA to perform the analysis described.
Lines 95-100: please provide more details about the surveys.
Line 103: did you really sun dried the samples? Why? Did that not decrease quality and quantity of DNA?
Line 104: how many tissue samples? Where were they collected?
Line 135 onward: did you respect the assumption needed for max-ent regarding sampling? Please provide more details on that
Line 181: Which samples did you use to calculate ADO and FA?
Results
Line 285: it is not clear why and how you finally selected the 7 msats from the fist 10 you selected before.
LINE NUMBERING IS MISSING FROM PAG 10 ONWARD
Pag 10 second paragraph: did you calculate Fst between the two “populations”? With such a igh number of admixed individuals, I would not be so sure that they are separate populations.
Pag 10 end of the second paragraph: which are the major cluster you identify in panel D of figure 3? (Gene.land results). This is not clear.
Figure 3: Panel c and d are not readable
Pag 11 second paragraphs: what do you mean with “novel” corridor? Was it not a corridor before? Or you mean newly identified? Which were the ones identified before? As you use the same expression several times throughout the text, you should be more careful in explaining what you mean.
Pag 11 Discussion. Check numbering of sections.
First paragraph: please provide scientific names for species.
Second paragraph: are you then sure that those are two populations? I would discuss this issue more carefully.
Reviewer 2 Report
The authors present an interesting study on genetic structure and landscape genetics of Capra sibirica in western India. The bright side of the manuscript is that to provide the important details on genetic structure of the species and practical details for the species’ conservation. However, some parts of the manuscript is not easy to understand and manuscript needs minor improvements. Therefore, I would like to make some suggestions to improve the quality of the paper as below:
Line 6: Corresponding Author sign should be "*"
Line 11: “Wildlife” should not be bold.
Line: 22 and 24 What do you mean by “good amount of gene flow” and “good gene-flow”? low, high or moderate? Please clarify.
Lines 24-25: sq. km -> sq.km
Lines 57-60: “In India, Siberian ibex is endemic to Ladakh and Lahaul & Spiti, Himachal Pradesh (hereafter; Himalayan ibex) which is a part of Trans-Himalayan region of India located at the western edge of Indian Himalayan region (IHR)”. Please reconsider these sentences. The species found also China, Mongolia, Russia, Kazakhstan, Kyrgyzstan, Kazakhstan, Tajikistan, Pakistan and Afghanistan. It distributes in Leh & Kargil districts of Ladakh and Lahaul-Spiti district of Himachal Pradesh in India but not endemic to this region.
Lines 68-69: “Genetic assessment of wildlife from this region is very finite and only a few species has been studied in details” -> “Genetic assessment of these species from this region is limited and only a few species were studied (references).”
Line 80: 86983km2 -> 86983 km2
Lines 75-93: The study area should in the Materials and methods section. “2.1. The study area”
1. Metarials and Methods -> 2. Metarials and Methods
Please check the numbers of the subsection of Materials and methods section
Lines 150-151: “highly correlated variables were omitted from the analysis” Which variables were omitted and which variables were used for the analysis? Please clarify. You may also add to supplementary file.
Line 254: 1. Results -> 3. Results
Lines 255-257: sq. km -> sq.km.
13311sq.km-> 13311 sq.km
Line 290: In Table 1, please delete the paragraph icon “¶”. Also please write as usual numbers instead of “5.2E-02”.
ADO -> Allel droup out (ADO). FA-> False allel (FA)
The authors added the membership probability plot to Figure 3B the DAPC plot. This is also informative and shows the group membership probability. However, the DAPC plot should also be showed. Authors can also add the DAPC plot to supplementary information.
1. Discussion -> 4. Discussion
“We found that genetic diversity estimates of Himalayan Ibex were comparable to the most ungulates such as Nubian Ibex, Korean goral, Hangul, Blue sheep” Please clarify or rephase the sentence.
climatic variable -> climatic variables
Limitations of the study should also be given.
Reviewer 3 Report
Jabin etc al describe a population genetic analysis of Himalayan ibex, using both mtDNA control region sequences and micro satellite alleles from over 100 individuals from the Indian trans Himalayan region. They report substructure within the populations, but with no signal of an isolation by distance model explaining these patterns. They also find a novel connectivity corridor within the study region.
This study is an important step.in better understanding the genetic landscape of the iconic mammal species Capra sibirica, still under studied at a population genomic level. This is contextualised within the paper by environmental factors and the looming threat of the climate crisis to sensitive species. As such as the study is both useful to regional and international researchers. The authors should be commended on their work and manuscript.
I found the study methodology to be reasonable. The SI appears appropriate although the text in SI Figure 1 does not seem to have composited correctly, please fix.
I have five minor comments for the paper.
1) please improve the labeling of subfigures. For example figure 1 has no A, B etc labels. Other figures do have sublabels but in the same font and text size of axis labels within the figure. Labels should be clear, distinct from other text (eg right parentheses) , and not too close to the image displayed.
2) Greater context could be given in the introduction and in particular the discussion, relating to the study species. How does genetic diversity scores compare with other siberian ibex populations, if these are available? How does the Himalayan ibex behaviour (migratory behaviour, dispersal tendencies) and ecological preference relate to the geographical and population structure observed? The abstract could also be more specific that the study is of Capra sibirica.
3) I am a little unclear as to the interpretation of the demographic history inferences, which I would not describe as "corroborated" (line 8 of the discussion, note the line numbering stops in the middle of the paper). The results conflict or are seemingly stable e.g. Fig 2, here even the possible recent decline gives confidence intervals that would fit with a stable population. The results are probably closer to not being able to reject a stable population size, at leat for the older times periods and likely as a consequence of limited power.
4) another pass should be made on English - it's rarely wrong per se but can be more clear. For example, the abstract in line 13 uses the word "ensemble" which is not typical, "co-analysis" may be more appropriate. On line 49, "across larger landscapes" rather than "in large landscape". There are several more examples, so I recommend the text be given another check for language.
5) will the sequences generated as part of the study be made available? I could not see an accession or link in the paper.
Round 2
Reviewer 1 Report
I have read the reivsed version of the manuscript. Most of my methodological concerns were fullfilled.
I still think that the text can be improved, espacilly punctuation and that introduction would benefit from some expansion.
I also have few other requests:
Line 60: provide an explanation of why you decide to call it Himalayan ibex.
Line 81: For coherence with the other districts, provide also the complete name for HP (is it district of Himal-Pradesh or Lahaul-Spiti (LS) district of Himachal Pradesh as mentioned in the next lines 89-90?). In line 89-90 the names do not correspond to what presented in the map and there are acronyms that are not provided in the map. Please be coherent here and throughout the text.
I would also change - with : to make it clearer that the following names refers to that specific district or place them in brackets. For example:
The study was carried out in the following areas. In Ladakh (LA): Leh district (Changthang, Sham and Nubra valley) and Kargil District (Drass and Zanskar); In Himachal Pradesh (HP): Lahaul and Spiti (LS) district. <-- or is it the other way around and Hymachal Pradesh is the distrct? I still could not understand.
In the rest of the text, either you refer to the areas with their names or with the acronyms you provide, there is no need to repeat both of them every time (e.g. 245, 246, etc..).
Figures: increase font size of labels, axis and legends of all figures. They are still not readable (for example fig. 4D but also B and C and all the maps).
